# Diagnostic Performance of Serum MicroRNAs for ST-Segment Elevation Myocardial Infarction in the Emergency Department

**DOI:** 10.3390/biomedicines11092422

**Published:** 2023-08-30

**Authors:** Brianda Amezcua-Guerra, Luis M. Amezcua-Castillo, Jazmín A. Guerra-López, Kietseé A. Díaz-Domínguez, José L. Sánchez-Gloria, Andrés Cruz-Melendez, Adrián Hernández-Díazcouder, Yaneli Juárez-Vicuña, Fausto Sánchez-Muñoz, Fengyang Huang, Claudia Tavera-Alonso, Malinalli Brianza-Padilla, Elvira Varela-López, Daniel Sierra-Lara, Alexandra Arias-Mendoza, Gabriela Fonseca-Camarillo, Ricardo Márquez-Velasco, Héctor González-Pacheco, Rashidi Springall, Luis M. Amezcua-Guerra

**Affiliations:** 1School of Medicine, Universidad Nacional Autónoma de México, Mexico City 04510, Mexico; sofia.amezcua.2003@gmail.com; 2Coronary Care Unit, Instituto Nacional de Cardiología Ignacio Chávez, Mexico City 14080, Mexico; lmamezcuac@gmail.com (L.M.A.-C.); danielsierralaram@gmail.com (D.S.-L.); aariasm@yahoo.com (A.A.-M.); hectorglezp@hotmail.com (H.G.-P.); 3Department of Immunology, Instituto Nacional de Cardiología Ignacio Chávez, Mexico City 14080, Mexico; jazminaguerra@gmail.com (J.A.G.-L.); april9d@gmail.com (K.A.D.-D.); luis_san29@hotmail.com (J.L.S.-G.); adrian.hernandez.diazc@hotmail.com (A.H.-D.); yaneli2608@hotmail.com (Y.J.-V.); fausto22@yahoo.com (F.S.-M.); maly.brianz@gmail.com (M.B.-P.); gabrielafaster@gmail.com (G.F.-C.); marquezric@hotmail.com (R.M.-V.); 4Department of Internal Medicine, Rush University Medical Center, Chicago, IL 60612, USA; 5Core Lab, Instituto Nacional de Cardiología Ignacio Chávez, Mexico City 14080, Mexico; andres.cruz.qfb@outlook.com (A.C.-M.); taveramuc@yahoo.com.mx (C.T.-A.); 6Research Laboratory of Obesity and Asthma, Hospital Infantil de Mexico Federico Gómez, Mexico City 06720, Mexico; f_y_huang@yahoo.com; 7Translational Research Unit UNAM–INC, Instituto Nacional de Cardiología Ignacio Chávez, Mexico City 14080, Mexico; varelopz@yahoo.com; 8Health Care Department, Universidad Autónoma Metropolitana-Xochimilco, Mexico City 14387, Mexico

**Keywords:** microRNA, miR-133b, miR-126, miR-155, STEMI, myocardial infarction, cardiovascular disease

## Abstract

Prompt diagnosis of ST-segment elevation myocardial infarction (STEMI) is essential for initiating timely treatment. MicroRNAs have recently emerged as biomarkers in cardiovascular diseases. This study aimed to evaluate the discriminatory capacity of serum microRNAs in identifying an ischemic origin in patients presenting with chest discomfort to the Emergency Department. The study included 98 participants (78 with STEMI and 20 with nonischemic chest discomfort). Significant differences in the expression levels of miR-133b, miR-126, and miR-155 (but not miR-1, miR-208, and miR-208b) were observed between groups. miR-133b and miR-155 exhibited 97% and 93% sensitivity in identifying STEMI patients, respectively. miR-126 demonstrated a specificity of 90% in identifying STEMI patients. No significant associations were found between microRNAs and occurrence of major adverse cardiovascular events (MACE). However, patients with MACE had higher levels of interleukin (IL)-15, IL-21, IFN-γ-induced protein-10, and N-terminal pro B-type natriuretic peptide compared to non-MACE patients. Overall, there were significant associations among the expression levels of microRNAs. However, microRNAs did not demonstrate associations with either inflammatory markers or cardiovascular risk scores. This study highlights the potential of microRNAs, particularly miR-133b and miR-126, as diagnostic biomarkers for distinguishing patients with STEMI from those presenting with nonischemic chest discomfort to the Emergency Department.

## 1. Introduction

Coronary artery disease (CAD) is the leading cause of mortality worldwide. In the United States, approximately 735,000 individuals experience a myocardial infarction (MI) each year. Among them, 525,000 have an initial MI, while 210,000 face recurring episodes [1]. Evidence indicates that 75% of acute coronary syndromes (ACS) occur from the rupture of atherosclerotic plaques, with the highest incidence observed in men aged 45 and above. Conversely, women exhibit an increased incidence beyond the age of 50, perhaps due to the protective role of female sex hormones. The diagnosis of MI relies upon a comprehensive assessment encompassing clinical symptoms, blood markers, and electrocardiographic findings [1,2]. Although cardiac troponins T (cTnT) and I (cTnI) stand as the standard biomarkers for diagnosis, elevated levels can also be associated with other conditions, including myocarditis, sepsis, chronic kidney disease, and pulmonary embolism. Consequently, the development of novel biomarkers to identify myocardial damage promptly and accurately is of utmost urgency. These biomarkers would not only aid in early diagnosis but also provide prognostic information and facilitate the timely initiation of appropriate treatment strategies for MI patients [3].

MicroRNAs (miRNAs) are a class of endogenous, non-coding, single-stranded RNAs consisting of approximately 20 nucleotides. These molecules exert their regulatory effects by negatively modulating gene expression at a post-transcriptional level, thereby influencing critical processes such as cell development, differentiation, proliferation, and metabolism [4]. Over the past few years, miRNAs have emerged as promising biomarkers due to their involvement in various pathophysiological mechanisms within the realm of cardiovascular diseases [3,5]. One of the key advantages of miRNAs as clinical biomarkers stems from their accessibility, as they can be readily obtained from various biological sources, including blood, urine, and tissue samples. The availability of feasible detection methods, such as quantitative polymerase chain reaction (qPCR) and next-generation sequencing, further supports the integration of miRNA measurements into routine clinical practice [3,4,5]. The multifaceted roles of miRNAs in disease processes, particularly within cardiovascular diseases, make them valuable candidates for diagnostic, prognostic, and therapeutic applications. By harnessing the unique properties of miRNAs, researchers and clinicians can potentially gain novel insights into disease mechanisms, improve early detection, enhance risk stratification, and develop targeted therapeutic interventions [3].

This study aimed to assess the discriminatory potential of several miRNAs in identifying the ischemic origin in patients presenting to the Emergency Department (ED) with chest discomfort suggestive of MI. Furthermore, we evaluated the predictive value of miRNAs for the in-hospital occurrence of major adverse cardiovascular events (MACE). Finally, the study sought to explore the relationship between miRNAs and inflammation-related molecules, such as cytokines and C-reactive protein (hsCRP), as well as their correlation with conventional MI biomarkers.

## 2. Materials and Methods

### 2.1. Study Design

A prospective, observational, and analytical study was conducted at the Instituto Nacional de Cardiología Ignacio Chávez in Mexico City, a university hospital devoted to cardiovascular diseases and allied conditions. Patients were consecutively recruited and were eligible for inclusion if they were admitted to the ED with symptoms suggestive of an ACS, including chest, upper extremity, mandibular, or epigastric discomfort. Patients who presented with ischemic equivalents such as dyspnea or fatigue were also considered. Patients were excluded if they had been receiving glucocorticoids within the last three months, had a cancer diagnosis, active infection, were pregnant, or declined to participate in the study.

Patients underwent a comprehensive evaluation by the medical staff in the ED and the diagnosis of MI was confirmed or ruled out based on standardized guidelines [2]. It is important to note that the researchers did not intervene in the diagnostic process and the need for cardiac catheterization and other therapeutic decisions rested solely on the treating physician. Patients who received a diagnosis of ST-segment elevation myocardial infarction (STEMI) served as the case group and were closely monitored for the occurrence of MACE throughout their hospitalization. Patients diagnosed with non-ST-elevation MI or unstable angina were excluded. Patients who presented with chest discomfort but were ruled out for an ischemic origin served as the control group.

Informed consent was obtained from the patients or their legal representatives, allowing the use of their blood samples and clinical data for research purposes. The study received approval from the local ethics committee (protocol number 21-1273). All procedures were conducted in accordance with the principles outlined in the Declaration of Helsinki, its annexes, and local regulations.

### 2.2. Clinical Assessment

Upon hospital admission, demographic, clinical, and laboratory data were collected for each patient. STEMI severity was evaluated using the Global Registry of Acute Coronary Events (GRACE) and the Thrombolysis in Myocardial Infarction (TIMI) risk scores at the time of admission to the Coronary Care Unit [6,7]. Information regarding the results of cardiac catheterization was obtained from the reports generated by the hemodynamicist responsible for performing the procedure. The occurrence of MACE was determined based on a composite outcome measure including acute heart failure, pulmonary edema, cardiogenic shock, angina, stroke, and death.

### 2.3. Laboratory Procedures

Upon hospital admission, prior to any medical intervention or pharmaceutical administration, venous blood samples were obtained from each participant and centrifuged (600× *g*, 15 min, 4 °C). The resulting sera were stored at −70 °C until use.

### 2.4. Quantification of Cytokines

To quantify the levels of various cytokines, a serum aliquot was carefully thawed following standardized procedures. The levels of interleukin (IL)-17F, IL-13, IL-15, IL-17A, IL-9, IL-1β, IL-33, IL-21, IL-4, IL-23, IL-5, IL-6, IL-17E, interferon-γ (IFN-γ), and tumor necrosis factor (TNF) were assessed using the Milliplex MAP human Th17 magnetic bead panel (Millipore; Burlington, MA, USA). Cytokines were analyzed using a Luminex 200 system (Luminex Corp; Austin, TX, USA), and the 3.1 “Xponent” software (Luminex Corp) was used for the data analyses.

IFN-γ-induced protein-10 (IP-10) was measured by enzyme-linked immunosorbent assay following the manufacturer’s protocol (R&D Systems; Minneapolis, MN, USA; range 7.8–500 pg/mL).

### 2.5. Quantification of MicroRNAs

To isolate RNA, a centrifugation step was conducted with 100 µL of serum at 10,000× *g* for 30 min at 4 °C. This process effectively removed cellular debris and allowed for the collection of the resulting supernatant. The RNA isolation procedure from the serum samples was accomplished using the serum/plasma mini kit from Qiagen (Qiagen; Hilden, Germany). During the RNA purification step, an equal amount of cel-miR-39 spike-in control (Qiagen) was added, and the obtained RNA was promptly converted into complementary DNA (cDNA).

To identify the presence of miRNAs, we used a two-step RT-qPCR method involving a specific RT-primer assay along with TaqMan probes. The miRNAs under analysis were miR-1 (assay ID: 002222), miR-208 (assay ID: 000511), miR-208b (assay ID: 002290), miR-133b (assay ID: 002247), miR-126 (assay ID: 002228), miR-155 (assay ID: 002623), and cel-miR-39 (assay ID: 000200) (Applied Biosystems; Foster City, CA, USA). For the RT reaction, 1.5 µL of the eluted RNA (obtained from a 14 µL sample) was used, and the TaqMan MicroRNA Reverse Transcription Kit (Applied Biosystems) was employed. The RT and PCR reactions were conducted using a CFX96 instrument (BioRad; Hercules, CA, USA). To normalize the relative concentrations of the miRNAs, the Ct values of cel-miR-39 were utilized. The calculations were performed using the 2^−ΔCt^ formula, and the resulting values were multiplied by 1000.

### 2.6. Statistics

Discrete variables were expressed as frequencies (percentages) and analyzed using the Fisher’s exact test. Continuous variables were described as medians (interquartile range) and analyzed using the Mann–Whitney U test. The associations between variables were evaluated using Spearman’s ρ coefficient, and these correlations were visually represented through a correlation matrix.

To assess the diagnostic potential of miRNAs, the area under the receiver operating characteristic curve (AUC-ROC) was calculated. Additionally, the optimal cut-off point for each miRNA was determined using Youden’s J index (J = sensitivity + specificity − 1). Confidence intervals at the 95% level were calculated.

All statistical analyses were two-tailed, and a significance threshold of *p* < 0.05 was set. The Social Science Statistics website (http://socscistatistics.com) and GraphPad Prism version 9.5.1 (GraphPad Software; La Jolla, CA, USA) were utilized for data calculations and visualization.

## 3. Results

### 3.1. Study Population

The study enrolled a total of 98 participants, consisting of 78 patients with STEMI and 20 individuals experiencing non-ischemic chest discomfort. The main clinical features are summarized in Table 1. Notably, none of the control subjects exhibited underlying conditions characterized by anxiety, and only one individual was identified as having experienced a panic attack. The prevailing diagnoses among the control group primarily comprised cases of costochondritis and non-specific musculoskeletal pain.

Among the STEMI patients, 88% were male, with a median age of 60 years (53–65). The prevalent comorbidities in STEMI patients included systemic hypertension (52%) and diabetes mellitus (39%), with 37% reporting a smoking habit. Table 2 summarizes their baseline and clinical characteristics.

### 3.2. MicroRNA Expression Differences

A first analysis revealed differences in miRNA expression between STEMI patients and individuals with non-ischemic chest pain (Figure 1). Specifically, miR-1 (16.2, 7.8–26.1 vs. 11.7, 8.9–14.8; *p* = 0.560), miR-208 (1.3, 0.4–2.3 vs. 0.6, 0.3–0.9; *p* = 0.068), miR-208b (0.8, 0.2–2.8 vs. 0.6, 0.1–1.9; *p* = 0.577), miR-133b (20.6, 9.2–41.5 vs. 10.0, 2.6–26.5; *p* = 0.014), miR-126 (75.6, 52.4–103.6 vs. 50.5, 40.1–59.3; *p* = 0.001), and miR-155 (55.0, 33.3–80.9 vs. 131.6, 38.7–38.7; *p* = 0.018) exhibited varying expression levels between STEMI patients and individuals with non-ischemic chest pain, respectively.

### 3.3. Predictive Capacity of MicroRNAs for STEMI Occurrence

Further investigation assessed the discriminatory capacity of significant miRNAs to identify STEMI occurrence (Figure 2). For miR-133b, an AUC-ROC of 0.67 (0.53–0.82; *p* = 0.015) was achieved, with a sensitivity of 97% and negative likelihood ratio of 0.06 at a cut-off point of >3.52 units. MiR-126 showed an AUC-ROC of 0.72 (0.61–0.84; *p* = 0.001), with a specificity of 90% and positive likelihood ratio of 5.6 at a cut-off point of >70.4 units. miR-155 exhibited an AUC-ROC of 0.67 (0.51–0.82; *p* = 0.019) with a sensitivity of 93% and negative likelihood ratio of 0.13 at a cut-off point of <147 units.

### 3.4. Major Adverse Cardiovascular Events and Related Factors

Among the STEMI patients, 17% (14 patients) experienced MACEs during follow-up. Notably, diabetes mellitus prevalence was higher in the MACE group (71% vs. 32%; *p* = 0.013). No significant differences in other comorbid conditions were observed (Table 2). Laboratory data, presented in Table 3, revealed that MACE patients had lower serum albumin levels (3.9, 3.3–4.2 vs. 4.2, 3.9–4.4; *p* = 0.030) and higher N-terminal pro B-type natriuretic peptide levels (NT-proBNP; 554, 217–2027 vs. 1936, 488–5000; *p* = 0.040) than non-MACE patients.

The clinical course and angiographic findings of STEMI patients are summarized in Table 4. Anti-ischemic treatment, primary percutaneous coronary intervention, and atherosclerotic burden were similar between groups. However, intra-aortic balloon pump support was more frequent in the MACE group (35% vs. 6%; *p* = 0.007). Acute heart failure (12 cases) and cardiogenic shock (4 cases) were the most common MACE. Additionally, two patients died during follow-up. MACE patients had longer hospital stays (9, 4–9 vs. 4, 2–6 days; *p* = 0.005). The predictive ability of miRNAs for MACE occurrence was assessed (Figure 3), showing no significant results. Similarly, no significant differences were found in miRNA levels according to the site of MI, assessed according to the predominantly involved coronary vessel.

### 3.5. Inflammatory Molecules and Major Adverse Cardiovascular Events

Serum levels of inflammatory molecules were measured in STEMI patients according to the occurrence of MACE (Table 5). While most cytokines exhibited similar levels, IL-15 (6.40, 5.64–8.68 vs. 4.88, 2.90–6.86; *p* = 0.041), IL-21 (6.07, 0.75–9.64 vs. 0.16, 0.08–4.88; *p* = 0.044), and IP-10 (125.73, 76.87–237.66 vs. 72.13, 48.20–99.74; *p* = 0.005) were significantly higher in MACE patients.

### 3.6. Association of MicroRNAs with Inflammatory Markers and Cardiovascular Risk Scores

In the final analysis, the correlations among miRNAs, inflammatory molecules, myocardial lysis markers, and cardiovascular risk scores were assessed. Notably, no significant associations were found between miRNAs and inflammatory markers or cardiovascular risk scores. Albumin levels negatively correlated with hsCRP and TNF, and positively correlated with IL-15 and IL-6. Other significant associations included IL-15 with the GRACE score, IL-13 with leukocytes, and cTnT with IL-6, hsCRP, and albumin (Figure 4).

## 4. Discussion

This study aimed to assess the involvement of specific miRNAs in MI. Our findings revealed that miRNA-133b and miRNA-126 exhibited high sensitivity in identifying the ischemic origin of patients presenting with chest discomfort upon admission to the ED. In addition, miRNA-155 demonstrated high specificity in distinguishing the ischemic origin in these patients. However, it is important to note that miRNAs did not demonstrate the ability to predict the severity or clinical course of patients with STEMI.

The primary finding of this study highlights the potential of certain miRNAs to effectively differentiate patients with STEMI from those presenting with non-ischemic chest discomfort. Currently, diagnosing MI in the ED poses several challenges. These challenges include reliance on the physician’s expertise and skills, as well as the accuracy of triage classification within the first minutes of a patient’s arrival. Compounding these difficulties are administrative issues and staffing shortages in the ED, which further hinder prompt and accurate diagnosis of MI [8]. The laboratory assessment for diagnosing MI presents certain challenges as well. When reviewing the electrocardiogram, the ability to identify specific abnormalities depends on the phase of ischemia the patient is experiencing. Additionally, early MI detection using markers such as creatine kinase-myocardial band (CK-MB) is limited due to the diffusion of these markers from the infarcted tissue into the systemic circulation [9]. CK-MB becomes detectable only after 4 to 6 h, while cTnT as well can be detected after 4 to 6 h and reaches its peak concentration around 10 h after the onset of symptoms [10]. Angiography, considered the gold standard for diagnosing CAD, is an invasive procedure associated with specific requirements and potential complications. It necessitates the expertise of a physician specialized in critical care and requires technical proficiency. As with any invasive procedure, there are inherent risks of complications such as bruising, nausea, pain, or reactions at the puncture site. Transitory decreases in renal function may also occur. The overall risk of major complications is less than 2%, which includes rare occurrences of anaphylactoid reactions and acute renal failure [11].

A few studies converge to lead plausibility to our findings, specifically pertaining to the substantial involvement of miRNAs in the initiation and progression of vascular thrombi. A seminal study showed that miR-133, along with its regulatory circuit involving the sirtuin-1, a nicotinamide adenine dinucleotide (NAD)-dependent histone deacetylase, is overexpressed in coronary thrombi [12]. This study also underscored the notable impact of miR-133 overexpression on the burden of coronary thrombi in patients with STEMI. In a related vein, the influence of inflammatory and prothrombotic content within the vascular thrombi has been highlighted in an additional study, which confirmed that thrombus aspiration as an adjunct to percutaneous coronary intervention, as opposed to thromboplasty without concomitant aspiration, exerts favorable effects on medium- and long-term outcomes in hyperglycemic patients with STEMI [13]. Our current findings complement these earlier results and collectively support the idea that miRNAs could function at different levels. This encompasses their role as epigenetic modulators of platelet activation and aggregation, along with their direct regulation of prothrombotic processes within the coronary microenvironment.

miR-133b, encoded by the MIR133b gene on chromosome 6, is categorized as one of the muscle-specific miRNAs. In recent studies, miR-133b has also been found to play a role in heart development and various cardiac-related disorders, including MI and cardiac hypertrophy. Notably, decreased levels of miR-133b have been associated with these conditions [14]. In a study conducted by Carè and colleagues, miR-133b demonstrated three significant targets: RhoA, Cdc42, and NELFA/Whsc2. These targets are known to be involved in myofibrillar rearrangement, which plays a critical role in cardiac hypertrophy. In the context of MI, miR-133b expression may be upregulated due to tissue damage. Conversely, in cases of non-ischemic chest discomfort, where no tissue damage is present, the expression of miR-133b tends to be lower or within normal ranges [15].

miR-155, encoded by the MIRHG155 gene, exhibits expression in various cell types, including B cells, T cells, and endothelial cells. Extensive evidence supports its involvement in hematopoiesis, inflammation, immunity, neoplasms, and cardiovascular diseases. Notably, recent research highlights the significance of miR-155 in the pathogenesis of autoimmune diseases [16,17]. The molecule is recognized as a critical modulator of immune responses, stimulating inflammatory pathways such as MAPK, insulin, Wnt, and MAPK/NF-κB [18]. In relation to CAD, previous studies have shown the downregulation of miR-155 in MI patients. This downregulation aligns with the notion that miR-155 may play a role in ameliorating cardiac remodeling and hypertrophy following MI. Consequently, it may contribute to a better post-MI heart functioning [19]. In an experimental study using a mouse model of MI, the inhibition of miR-155 was found to improve cardiac dysfunction. This study demonstrated that miR-155 played a role in increasing the levels of IL-1β and matrix metalloprotease-7 (MMP7) in proinflammatory macrophages. This increase subsequently led to the degradation of connexin-43 in cardiomyocytes through paracrine signaling. The degradation of connexin-43 is known to have a detrimental effect on cardiac function [20]. Another experimental study revealed that miR-155 levels were upregulated after 7 days of MI. A miR-155 mimic inhibited cardiac fibroblast proliferation by downregulating Son of Sevenless 1 (SOS1) expression and promoted inflammation by decreasing Suppressor of Cytokine Signaling 1 (SOCS1) expression. Furthermore, the levels of IL-1β, IL-6, TNF, and chemokines were significantly decreased [21]. These findings suggest that the downregulation of miR-155 accelerates the process of fibroblast proliferation and reduces inflammation in CAD.

miR-126 has been extensively studied in stroke and cardiovascular diseases due to its involvement in promoting angiogenesis, facilitating vascular remodeling, and reducing fibrosis. In an experimental model, it was demonstrated that miR-126 was highly expressed under ischemic and hypoxic conditions and played a crucial role in promoting angiogenesis during MI by upregulating the expression of vascular endothelial growth factor (VEGF) and CD34 [22]. Considering the expression of this miRNA in hypoxic situations, it is reasonable to propose that miR-126 could serve as a diagnostic tool for MI. Another experimental study demonstrated an upregulation of miR-126 expression levels, consistent with the findings in our study. Compared to the control group, the MI group exhibited evident myocardial fibrosis and collagen aggregation, emphasizing the significant role of miR-126 in the remodeling process and angiogenesis following MI [23].

Another interesting finding of this study is that miRNAs do not appear to exert a decisive influence on the progression and severity of STEMI. Conversely, patients who experienced MACEs demonstrated considerably elevated levels of NT-proBNP, IL-15, IL-21, and IP-10, while exhibiting lower levels of albumin upon hospital admission. These findings further substantiate the notion that inflammation and hemodynamic alterations stemming from acute heart failure are the principal factors governing the clinical course of MI [24]. In the setting of myocardial ischemia, when cardiomyocytes release damage-associated molecular patterns (DAMPs) and reactive oxygen species (ROS), a pro-inflammatory response ensues. This inflammatory milieu is characterized by an abundance of interleukins (IL-1α, IL-1β, IL-6, IL-18) and chemokines (CCL2, CCL18), and triggers the activation of the complement pathway, as well as the production of acute-phase reactants such as hsCRP [25]. In our study, we observed that patients who experienced MACE exhibited reduced albumin levels and elevated NT-proBNP levels. A previous investigation by González-Pacheco and colleagues revealed that MI patients with hypoalbuminemia displayed lower left ventricular ejection fraction and higher mortality, indicating that hypoalbuminemia may contribute to the progression of heart failure by promoting myocardial edema and facilitate pulmonary edema by reducing serum oncotic pressure [26]. In the context of hemodynamics, ventricular myocytes release NT-proBNP in response to stretching and volume overload. This biomarker serves as an important indicator to assess the magnitude of infarcted tissue and left ventricular systolic dysfunction [27]. In a study conducted by Radosavljevic-Radovanovic and colleagues, it was found that MI patients with elevated NT-proBNP values had a three-fold higher risk of developing incident heart failure [28].

While it remains evident that miRNAs may not exert definitive influence on the acute phase severity of STEMI, the potential regulatory role they assume in the context of postinfarction reparative processes, primarily through myofibroblast phenoconversion, warrants exploration. This premise finds support in an experimental investigation conducted in a mouse model, wherein a notable upregulation of miR-195 was observed within cardiosomes and isolated fibroblasts after MI [29]. This study revealed that the activation of primary cardiac fibroblasts occurred when exposed to cardiosomes sourced from ischemic cardiomyocytes and when subjected to post-MI cardiomyocyte-conditioned medium. Such findings suggest a plausible involvement of miRNA-mediated mechanisms in orchestrating post-MI reparative pathways, facilitated by the transformation of fibroblasts into myofibroblasts. This orchestrated phenoconversion could potentially exert downstream effects on the process of post-MI remodeling, consequently impacting the trajectory toward heart failure and other deleterious sequelae over the extended course of disease progression.

It is important to acknowledge that our study possesses several limitations. Firstly, it is a single-center study conducted exclusively at a cardiovascular disease management facility. Secondly, the duration of patient recruitment was relatively short, and the sample size was not extensive. Thirdly, we did not conduct serial measurements of circulating miRNAs. Fourthly, disparities in age and the prevalence of comorbidities between the patient group and the control individuals could introduce confounding variables that may influence the observed results. Lastly, the Mexican population included in our study represents a mixture of European, Native American, and African ancestry (52%, 44%, and 4%, respectively), which restricts the generalizability of our findings to other populations [30]. Despite these limitations, it is worth noting that all patients included in our study received optimal reperfusion therapy, including early percutaneous coronary intervention. Moreover, blood samples were collected at the time of admission to the ED, before any pharmacological or interventional treatment was administered to the patients.

## 5. Conclusions

Our study highlights the promising diagnostic potential of miR-133b, miR-155, and miR-126 in distinguishing patients with STEMI from those with nonischemic chest discomfort in the ED. Implementing these biomarkers has the potential to yield improved patient outcomes and precision in the management of acute cardiovascular conditions.

## Figures and Tables

**Figure 1 biomedicines-11-02422-f001:**
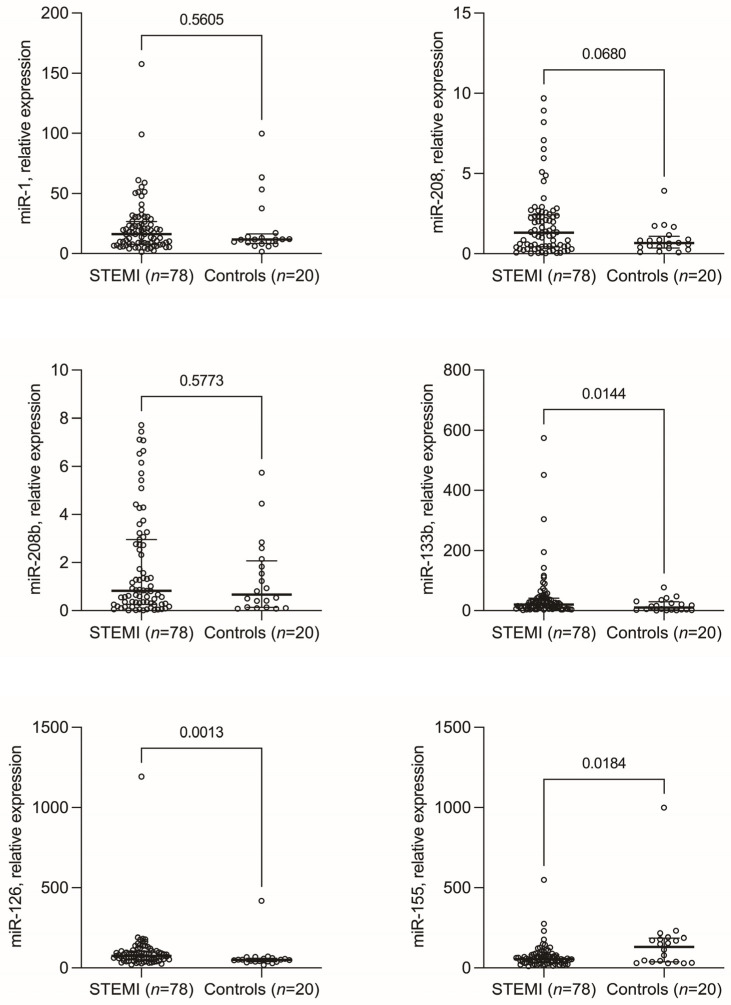
Median serum levels of microRNAs in patients with ST-segment elevation myocardial infarction (STEMI) and individuals with non-ischemic chest discomfort (Controls).

**Figure 2 biomedicines-11-02422-f002:**
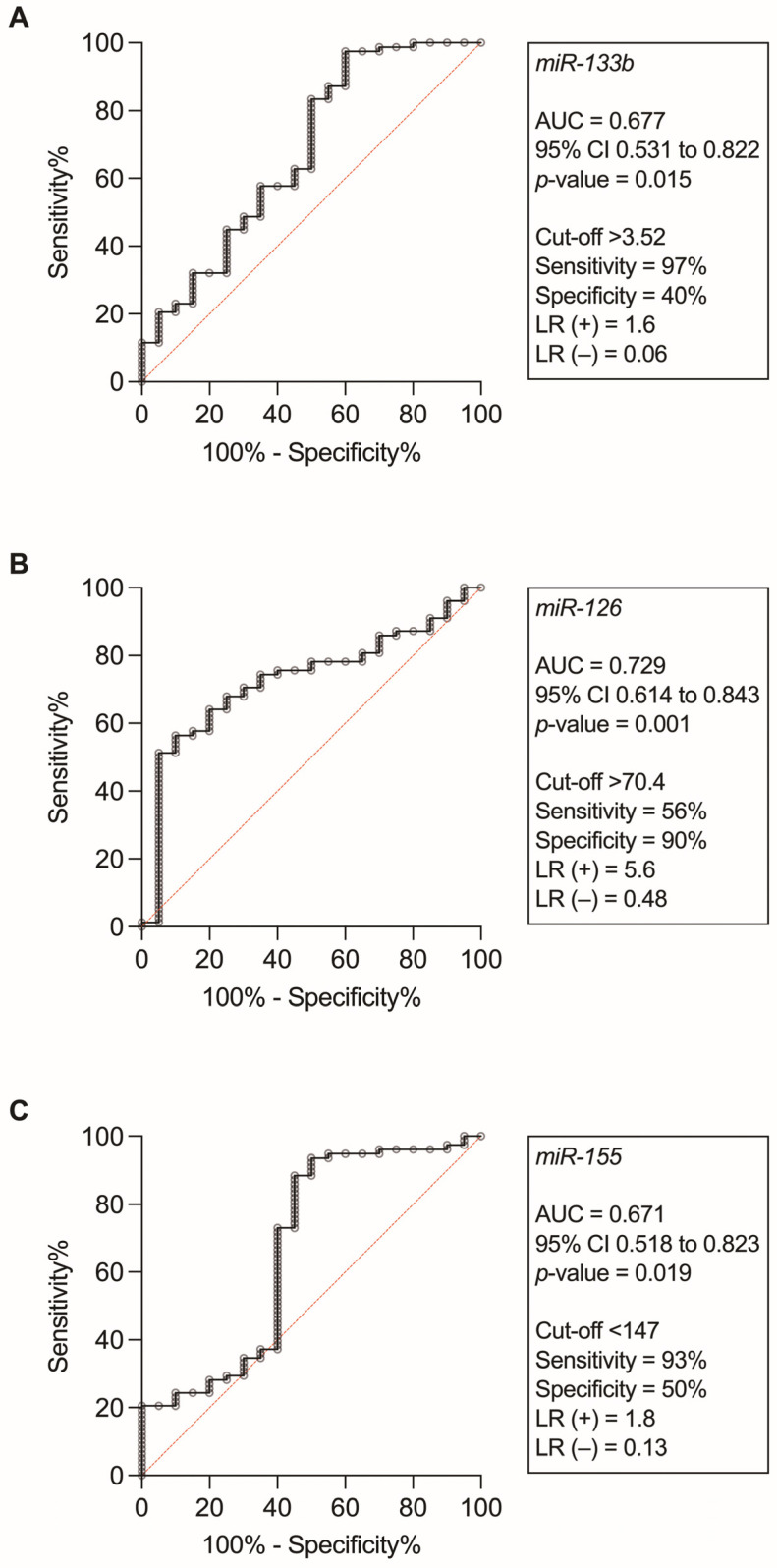
The area under the receiver operating characteristic curve (AUC) for miR-133b (panel **A**), miR-126 (panel **B**), and miR-155 (panel **C**) and their performance to identify individuals with ST-segment elevation myocardial infarction. LR denotes likelihood ratio.

**Figure 3 biomedicines-11-02422-f003:**
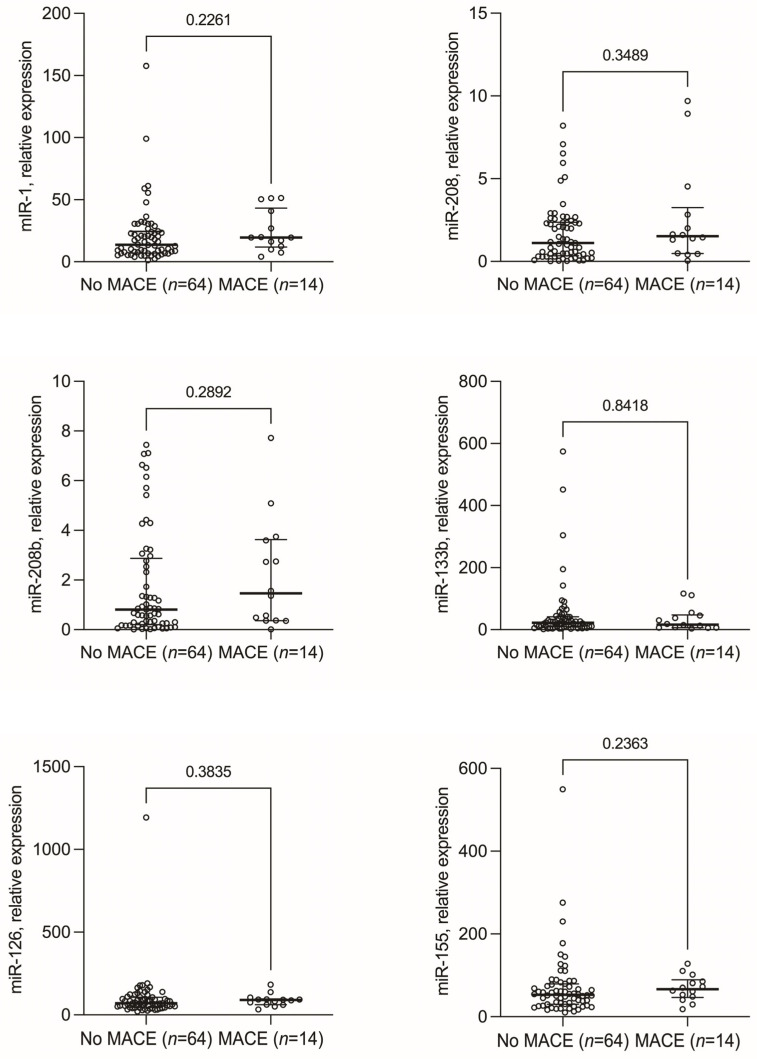
Median serum levels of microRNAs in patients with ST-segment elevation myocardial infarction according to the in-hospital occurrence of mayor adverse cardiovascular events (MACE).

**Figure 4 biomedicines-11-02422-f004:**
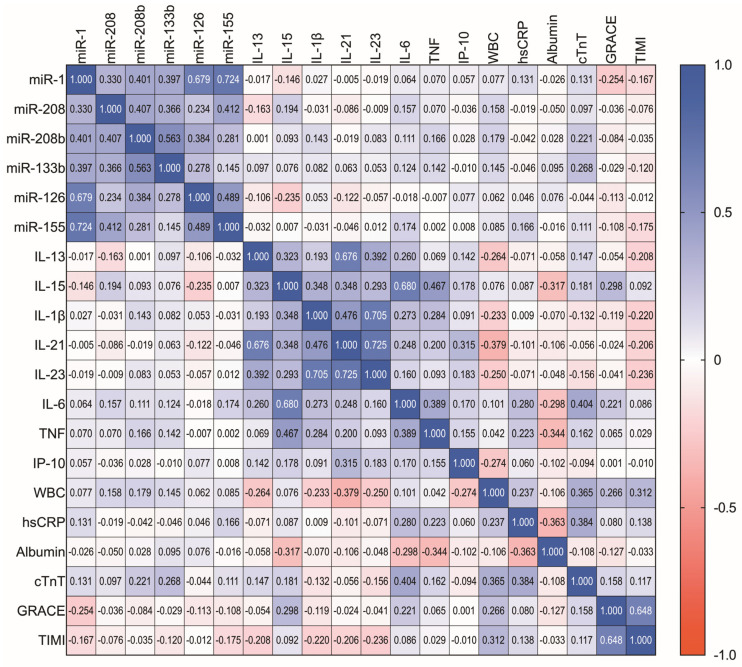
Correlation matrix illustrating the correlations among key inflammatory molecules, markers of myocardial injury, serum miRNA levels, and cardiovascular risk scores. The numerical value in each cell represents the Spearman’s correlation coefficient for the corresponding elements at the intersection. To facilitate visual interpretation, the color intensity reflects the strength of the association, with blue indicating positive correlations and red indicating negative correlations.

**Table 1 biomedicines-11-02422-t001:** Main clinical features of study participants.

	STEMI Patients (*n* = 78)	Non-Ischemic Chest Discomfort (*n* = 20)	*p*-Value
Age, years	60 (53–65)	36 (34–45)	**<0.001**
Male, n (%)	69 (88)	16 (80)	0.458
Heart rate, beats/min	80 (68–95)	70 (66–77)	**0.045**
Respiratory rate, breaths/min	18 (18–20)	18 (16–18)	**0.012**
Systolic blood pressure, mmHg	131 (118–148)	130 (120–141)	0.823
Diastolic blood pressure, mmHg	80 (70–90)	70 (70–88)	0.390
Body mass index, kg/m^2^	27.0 (24.5–29.7)	27.3 (24.2–29.6)	0.816
Coexisting conditions, n (%)			
Hypertension	41 (52)	3 (15)	**0.002**
Diabetes mellitus	31 (39)	1 (5)	**0.002**
Dyslipidemia	22 (28)	2 (10)	0.143
Current smoker	29 (37)	10 (50)	0.316
Previous MI	11 (14)	0	0.113
Autoimmune disease	1 (1)	0	>0.999
Leukocytes ×10^3^ per mm^3^	11.4 (9.4–14.0)	11.0 (9.8–11.8)	**<0.001**
Platelets ×10^3^ per mm^3^	230 (198–282)	230 (200–255)	0.656
Creatine kinase, U/L	825 (262–1818)	151 (81–214)	**<0.001**
CK-MB, U/L	71 (11–167)	1.2 (0.8–1.9)	**<0.001**
cTnT, ng/mL	7574 (844–100,000)	0.1 (0.1–0.1)	**<0.001**

All data are presented as median (interquartile range) unless otherwise specified. Significant *p*-values are in bold. Definitions: CK-MB, MB isoenzyme of creatine kinase; cTnT, cardiac troponin T. MI, myocardial infarction; STEMI, ST-segment elevation myocardial infarction.

**Table 2 biomedicines-11-02422-t002:** Baseline and clinical characteristics of patients with ST-segment elevation myocardial infarction at hospital admission.

	All Patients (*n* = 78)	Non-MACE (*n* = 64)	MACE (*n* = 14)	*p*-Value
Age, years	60 (53–65)	60 (53–65)	61 (54–66)	0.659
Male, n (%)	69 (88)	56 (87)	13 (92)	>0.999
Heart rate, beats/min	80 (68–95)	80 (65–95)	86 (74–93)	0.555
Respiratory rate, breaths/min	18 (18–20)	18 (18–20)	19 (17–23)	0.502
Systolic blood pressure, mm Hg	131 (118–148)	132 (120–149)	129 (118–138)	0.561
Diastolic blood pressure, mm Hg	80 (70–90)	80 (70–90)	79 (69–90)	0.984
SaO_2_, % at room air	95 (93–96)	95 (93–96)	95 (94–97)	0.242
Body mass index, kg/m^2^	27.0 (24.5–29.7)	27.3 (24.3–29.8)	26.5 (25.1–28.3)	0.857
Coexisting conditions, n (%)				
Hypertension	41 (52)	35 (54)	6 (42)	0.556
Diabetes mellitus	31 (39)	21 (32)	10 (71)	**0.013**
Dyslipidemia	22 (28)	20 (31)	2 (14)	0.326
Current smoker	29 (37)	24 (37)	5 (35)	>0.999
Previous MI	11 (14)	8 (12)	3 (21)	0.405
Previous stroke	1 (1)	1 (1)	0 (0)	>0.999
Autoimmune disease	1 (1)	0	1 (7)	0.179
Killip classification	1 (1–2)	1 (1–2)	1.5 (1–2)	0.207
GRACE risk score	112 (93–134)	111 (91–133)	125 (98–143)	0.262
TIMI risk score	4 (2–5)	4 (2–5)	5 (3–6)	0.109

All data are presented as median (interquartile range) unless otherwise specified. Significant *p*-value is in bold. Definitions: GRACE, Global Registry of Acute Coronary Events risk score; MACE, major adverse cardiovascular events; MI, myocardial infarction; SaO_2_, oxygen saturation by pulse oximetry; TIMI, Thrombolysis in Myocardial Infarction score.

**Table 3 biomedicines-11-02422-t003:** Main laboratory findings at hospital admission.

	All Patients (*n* = 78)	Non-MACE (*n* = 64)	MACE (*n* = 14)	*p*-Value
Leukocytes ×10^3^ per mm^3^	11.4 (9.4–14.0)	11.4 (9.3–14.1)	12.1 (10.4–14.0)	0.548
Neutrophils ×10^3^ per mm^3^	9.4 (7.5–11.9)	9.3 (7.0–11.1)	10.5 (7.9–12.5)	0.337
Lymphocytes ×10^3^ per mm^3^	1.3 (1.0–1.7)	1.3 (1.0–1.7)	1.2 (1.0–1.4)	0.400
Platelets ×10^3^ per mm^3^	230 (198–282)	229 (198–283)	238 (201–277)	0.794
Hemoglobin, g/dL	15.5 (15.6–16.5)	15.7 (14.7–16.6)	15.1 (13.9–15.5)	0.384
Albumin, g/dL	4.2 (3.8–4.4)	4.2 (3.9–4.4)	3.9 (3.3–4.2)	**0.030**
Serum creatinine, mg/dL	1.1 (0.8–1.3)	1.1 (0.8–1.2)	1.1 (1.0–1.6)	0.091
cTnT, ng/mL	7574 (844–100,000)	8792 (776–100,000)	6576 (1152–12,952)	0.756
Creatine kinase, U/L	825 (262–1818)	825 (264–1868)	712.55 (227–1318)	0.589
CK-MB, U/L	71 (11–167)	71 (11–161)	73 (13–190)	0.764
Lactic dehydrogenase, U/L	315 (201–695)	305 (200–699)	369 (274–513)	0.992
C-reactive protein, mg/L	5.0 (2.2–16.6)	4.7 (1.9–12.3)	6.7 (5.1–34.6)	0.161
NT-proBNP, ng/L	646 (256–2501)	554 (217–2027)	1936 (488–5000)	**0.040**

All data are presented as median (interquartile range). Significant *p*-values are in bold. Definitions: CK-MB, MB isoenzyme of creatine kinase; cTnT, cardiac troponin T; MACE, major adverse cardiovascular events; NT-proBNP, N-terminal pro B-type natriuretic peptide.

**Table 4 biomedicines-11-02422-t004:** In-hospital treatment and major clinical outcomes in patients with ST-segment elevation myocardial infarction.

	All Patients (*n* = 78)	Non-MACE (*n* = 64)	MACE (*n* = 14)	*p*-Value
Drugs use, n (%)				
Acetylsalicylic acid	76 (97)	62 (96)	14 (100)	>0.999
P2Y12 inhibitors	78 (100)	64 (100)	14 (100)	>0.999
Statins	76 (97)	63 (98)	13 (92)	0.328
RAAS inhibitors	72 (92)	59 (92)	13 (92)	>0.999
Fibrinolytic therapy	17 (21)	16 (25)	1 (7)	0.281
Primary PCI	72 (92)	59 (92)	13 (92)	>0.999
Left main trunk	5 (6)	4 (6)	1 (7)	>0.999
Left anterior descending artery	54 (69)	45 (70)	9 (64)	0.751
Circumflex artery	29 (37)	23 (35)	6 (42)	0.761
Right coronary artery	43 (55)	34 (53)	9 (64)	0.558
Three-vascular disease	15 (19)	13 (20)	2 (14)	>0.999
Invasive mechanical ventilation, *n* (%)	2 (2)	1 (1)	1 (7)	0.328
Intra-aortic balloon pump, *n* (%)	9 (11)	4 (6)	5 (35)	**0.007**
Major clinical outcomes, *n* (%)				
Acute heart failure	12 (15)	-	12 (85)	**-**
Pulmonary edema	3 (3)	-	3 (21)	**-**
Cardiogenic shock	4 (5)	-	4 (28)	**-**
Death	2 (2)	-	2 (14)	**-**
Days of hospital stay, median (IQR)	4 (2–7)	4 (2–6)	9 (4–9)	**0.005**

Significant *p*-values are in bold. Definitions: RAAS, renin–angiotensin–aldosterone system; PCI, percutaneous coronary intervention; MACE, major adverse cardiovascular events; IQR, interquartile range.

**Table 5 biomedicines-11-02422-t005:** Serum cytokine levels in patients with ST-segment elevation myocardial infarction according to the in-hospital occurrence of major adverse cardiovascular events (MACE).

	Non-MACE (*n* = 78)	MACE (*n* = 14)	*p*-Value
IL-17F, ng/mL	0.00023 (0.00023–0.00023)	0.00023 (0.00023–0.00023)	0.610
IFN-γ, pg/mL	0.26 (0.26–0.26)	0.26 (0.26–0.26)	0.250
IL-13, pg/mL	18.85 (5.12–32.86)	19.73 (11.66–34.45)	0.541
IL-15, pg/mL	4.88 (2.90–6.86)	6.40 (5.64–8.68)	**0.041**
IL-17A, pg/mL	2.64 (2.64–2.64)	2.64 (2.64–2.64)	0.681
IL-9, pg/mL	0.26 (0.26–0.26)	0.26 (0.26–0.26)	0.284
IL-1β, pg/mL	18.85 (5.12–32.86)	0.54 (0.21–1.65)	0.186
IL-33, pg/mL	0.0008 (0.0008–0.0008)	0.0008 (0.0008–0.0008)	0.186
IL-21, pg/mL	0.16 (0.08–4.88)	6.07 (0.75–9.64)	**0.044**
IL-4, ng/mL	0.0031 (0.0031–0.0031)	0.0031 (0.0031–0.0031)	0.952
IL-23, ng/mL	0.36 (0.12–0.78)	0.67 (0.36–6.48)	0.107
IL-5, pg/mL	0.28 (0.28–0.28)	0.28 (0.28–0.28)	0.992
IL-6, pg/mL	8.73 (2.30–19.23)	19.66 (6.77–32.09)	0.061
IL-17E, ng/mL	0.36 (0.36–0.36)	0.36 (0.36–0.36)	0.992
TNF, pg/mL	7.87 (4.79–10.49)	11.59 (6.73–17.70)	0.071
IP-10, pg/mL	72.13 (48.20–99.74)	125.73 (76.87–237.66)	**0.005**

All data are presented as median (interquartile range). Significant *p*-values are in bold.

## Data Availability

Research data are available upon reasonable request.

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
