# Peer review of "Diagnostic Performance of Serum MicroRNAs for ST-Segment Elevation Myocardial Infarction in the Emergency Department"

_biomedicines, 2023, doi:10.3390/biomedicines11092422_

Round 1

Reviewer 1 Report

This manuscript examines the discriminatory potential of serum miRNAs in identifying an ischemic origin among patients presenting with chest discomfort suggestive of MI. The authors found that miR-133b, miR-155, and miR-126 exhibit promise as diagnostic biomarkers, suggesting a tangible opportunity for enhancing diagnostic accuracy. Overall, the study is well-performed with clear results. However, the following concerns need to be addressed.

1.     The logical flow of the results section needs to be enhanced. The authors should reinforce the connectivity between different segments. The inclusion of transitional sentences would effectively guide readers through the data analysis. Additionally, it is advisable to distinctly articulate the study's conclusion, solidifying the key takeaways drawn from the findings.

2.     In Figures 1 and 3, it would be better to incorporate the exact number of biological replicates (n) within the figure legends.

Author Response

I would like to express my sincere gratitude for the thoughtful and constructive feedback provided by the reviewers on my manuscript titled "Diagnostic performance of serum microRNAs for ST-segment elevation myocardial infarction in the Emergency Department", manuscript ID: biomedicines-2550884. Their valuable comments and suggestions have significantly enriched the quality, content, and presentation of the manuscript.

In accordance with the Editor's instructions, I have addressed each of the reviewer's comments and have provided a point-by-point response below:

  1. The logical flow of the results section needs to be enhanced. The authors should reinforce the connectivity between different segments. The inclusion of transitional sentences would effectively guide readers through the data analysis. Additionally, it is advisable to distinctly articulate the study's conclusion, solidifying the key takeaways drawn from the findings.

Response: We appreciate the reviewer's valuable input in improving the logical flow of the results section. We have taken significant steps to enhance the coherence and organization of the results by incorporating transitional sentences to facilitate a smooth transition between different segments, in addition to include subheadings for each segment. This improvement aims to guide readers through the data analysis more effectively. Furthermore, we have worked on articulating a clearer and more distinct study conclusion that solidifies the key insights derived from our findings. The conclusion now reads as follows: "Our study highlights the promising diagnostic potential of miR-133b, miR-155, and miR-126 in distinguishing patients with STEMI from those with nonischemic chest discomfort in the ED. Implementing these biomarkers has the potential to yield improved patient outcomes and precision in the management of acute cardiovascular conditions."

  1. In Figures 1 and 3, it would be better to incorporate the exact number of biological replicates (n) within the figure legends.

Response: We sincerely acknowledge the reviewer's suggestion regarding Figures 1 and 3. Based on this feedback, we have now included the exact number of biological replicates (n) within the figure legends for both Figures 1 and 3. This enhancement ensures transparency and clarity in presenting the experimental design.

I'm confident that these revisions have substantially enhanced the clarity, coherence, and overall quality of the manuscript. Once again, I would like to extend my appreciation to the reviewers and the editorial team for their diligence and dedication in evaluating the manuscript.

Reviewer 2 Report

The authors provided a manuscript on a interesting research. They look to a possible role of miRNA in diagnostic and prognostic evaluation of myocardial infarction. I really appreciate the aim and support the authors to continue in this field.

However there are some issue in this manuscript that need to be addressed

1-  I find no description of control group. Please provide information on this group like those provided in table 1 for study group.

2- there is no information on site of myocardial infarction. Is there a possible difference in the miRNA expression according to the site of STEMI?

3- The figure 4 is interesting but pretty obscure in meaning. I guess that the strong coloured box are the most significant. However, there was no indication in statistical methods about this point. This is not secondary point, because the strength in correlation may not be related to a significant one, due the small number of subject enrolled.

4- According to previous issue, the authors stated "Notably, albumin levels exhibited a negative correlation with hsCRP (r -0.36), TNF (r -0.34), and a positive correlation with IL-15 (r 0.08), and IL-6 (r 0.28). Other significant associations included IL-15 with the GRACE score (r 0.29), IL-13 with leukocytes (r -0.26), and cTnT with IL-6 (r 0.40), hsCRP (r 0.28), and albumin (r -0.29)". First, I wondering to understand how Albumin may significantly correlate to IL-15. Secondly, I did not understand why the authors indicated these correlations due the entire manuscript and research is focus on miRNA.

5- The two interesting miR-126 and miR-155 the authors find related to CAD. These miR result involved in atherogenesis by promoting inflammatory response, degradation of lipoproteins and vascular inflammation. It has been also described in Atrial Fibrillation (https://doi.org/10.3390/ijms24065307). Atrial fibrillation as well as Arrhythmias are often a major complication of myocardial infarction. Is there any new atrial fibrillation observed? Are there any other arrhythmia described?

6- What the authors identified as MACE? please specify

7- there are many miRNA already evaluated in CAD (https://doi.org/10.3390/jcdd8020022) or acute heart failure (10.1002/ejhf.495), but none appear evaluated by the authors. Why they did this choice?

Author Response

I would like to express my sincere gratitude for the thoughtful and constructive feedback provided by the reviewers on my manuscript titled "Diagnostic performance of serum microRNAs for ST-segment elevation myocardial infarction in the Emergency Department", manuscript ID: biomedicines-2550884. Their valuable comments and suggestions have significantly enriched the quality, content, and presentation of the manuscript.

In accordance with the Editor's instructions, I have addressed each of the reviewer's comments and have provided a point-by-point response below:

Reviewer 2:

1- I find no description of control group. Please provide information on this group like those provided in table 1 for study group.

Answer = We agree with the reviewer. We have included the main clinical and laboratory characteristics of individuals with non-ischemic chest pain (control group). We have done it in a paragraph of the "Results" section so as not to add one more column to Table 1, since it would deviate from the format accepted by the journal. In short, we add the following paragraph: “In the non-ischemic chest discomfort group, 80% were male, with a median age of 36 years (34-45). Vital signs revealed a heart rate of 70 beats per minute and a respiratory rate of 18 breaths per minute. Systolic blood pressure was 130 mm Hg, while diastolic blood pressure was 70 mm Hg. Preexisting medical conditions included systemic hypertension (15%), diabetes mellitus (5%), dyslipidemia (10%), and smoking history (50%). Biochemical assessments indicated a median creatine kinase concentration of 151 U/L (81-214), and CK-MB levels were at 1.2 U/L (0.8-1.9). Notably, none of the patients dis-played cTnT levels exceeding reference values”.

2- there is no information on site of myocardial infarction. Is there a possible difference in the miRNA expression according to the site of STEMI?

Answer = Indeed, that is a limitation of our database. However, we analyzed the relative expression levels of each miRNA by grouping the STEMI patients according to the artery responsible for the infarct, comparing those who had occlusion of the left anterior descending artery or the coronary trunk against those who did not have damage in this location. This was our best approach in trying to answer the question about the site of the infarct. As now stated in the manuscript, we found no differences. The following paragraph was added: “The predictive ability of miRNAs for MACE occurrence was assessed (Figure 3), showing no significant results. Similarly, no significant differences were found in miRNA levels according to the site of MI, assessed according to the predominantly involved coronary vessel (data not shown)”.

3- The figure 4 is interesting but pretty obscure in meaning. I guess that the strong coloured box are the most significant. However, there was no indication in statistical methods about this point. This is not secondary point, because the strength in correlation may not be related to a significant one, due the small number of subject enrolled.

Answer = We have modified the description of the statistical analysis as follows: “The associations between variables were evaluated using the Spearman’s rho coefficient, and these correlations were visually represented through a correlation matrix”.

Additionally, the respective Figure legend has been modified as follows: “Figure 4. Correlation matrix illustrating the correlations among key inflammatory molecules, markers of myocardial injury, serum miRNA levels, and cardiovascular risk scores. The numerical value in each cell represents the Spearman’s correlation coefficient for the corresponding elements at the intersection. To facilitate visual interpretation, the color intensity reflects the strength of the association, with blue indicating positive correlations and red indicating negative correlations”.

4- According to previous issue, the authors stated "Notably, albumin levels exhibited a negative correlation with hsCRP (r -0.36), TNF (r -0.34), and a positive correlation with IL-15 (r 0.08), and IL-6 (r 0.28). Other significant associations included IL-15 with the GRACE score (r 0.29), IL-13 with leukocytes (r -0.26), and cTnT with IL-6 (r 0.40), hsCRP (r 0.28), and albumin (r -0.29)". First, I wondering to understand how Albumin may significantly correlate to IL-15. Secondly, I did not understand why the authors indicated these correlations due the entire manuscript and research is focus on miRNA.

Answer = An interesting observation. We have discreetly modified the wording of that paragraph to avoid potential confusion. Regarding your questions, although there are no studies on the association between albumin and IL-15 in STEMI, this cytokine is well characterized in acute inflammatory processes. Specifically, IL-15 has been shown to be a reliable marker of cytokine storm, and its levels predict mortality in COVID-19, the paradigm of a hyperinflammatory state. Furthermore, the IL-15/albumin ratio has been explored as a marker that predicts mortality after SARS-CoV-2 infection.

Although it is true that the study focuses on miRNA, we sought to give it some functional representation in the form of regulation of inflammatory responses or myocardial damage, hence we have measured soluble inflammatory mediators, myocardial lysis enzymes, and cardiovascular risk scores.

5- The two interesting miR-126 and miR-155 the authors find related to CAD. These miR result involved in atherogenesis by promoting inflammatory response, degradation of lipoproteins and vascular inflammation. It has been also described in Atrial Fibrillation (https://doi.org/10.3390/ijms24065307). Atrial fibrillation as well as Arrhythmias are often a major complication of myocardial infarction. Is there any new atrial fibrillation observed? Are there any other arrhythmia described?

Answer = Unfortunately, we do not consider assessing the occurrence of atrial fibrillation or other arrhythmia during the hospital stay of patients, and the recording of these events in the electronic clinical record is not reliable enough to assess whether they actually occur. We took this interesting observation into account to propose a future research study.

6- What the authors identified as MACE? please specify

Answer = Clinical events that we consider to be MACE are described in section “2.2. Clinical assessment”. There, it reads as follows: “The occurrence of MACE was determined based on a composite outcome measure including acute heart failure, pulmonary edema, cardiogenic shock, angina, stroke, and death”.

7- there are many miRNA already evaluated in CAD (https://doi.org/10.3390/jcdd8020022) or acute heart failure (10.1002/ejhf.495), but none appear evaluated by the authors. Why they did this choice?

Answer = Indeed, there are multiple miRNAs that have already been evaluated in CAD. In this sense, all the miRNAs that we included in our analysis have already been studied in the context of the MI/ASCVD. Furthermore, these were the “cardiomiRNAs” that we had available in our research arsenal and, unfortunately, we no longer had the financial resources to acquire new primers. Hence, our selection of miRNAs was driven by both availability and bibliographic support for functioning as cardiomiRNAs.

I'm confident that these revisions have substantially enhanced the clarity, coherence, and overall quality of the manuscript. Once again, I would like to extend my appreciation to the reviewers and the editorial team for their diligence and dedication in evaluating the manuscript.

Round 2

Reviewer 2 Report

The authors provider a revised version of the manuscript. I appreciate their efforts in upgrading the main text and figure caption. 

I agree with most of answers/explanations. I still not appreciate the control group description. There is no standard deviation or inter quartile range of systolic, diastolic blood pressure e heart rate. Some other descriptive data are not provided. In example no bmi was given. Finally there is no comparison between control and study group. I suggest a supplementary table that include all these data. 

A part this the authors addressed the issues required

Author Response

Dear Reviewer,

Following your recommendations, we have incorporated all the clinical and laboratory data pertaining to the individuals comprising the non-ischemic chest discomfort control group. These data sets have been organized within an independent Table (Table 1). A comparative analysis has been performed, juxtaposing these data against the corresponding values presented by STEMI patients. 

Moreover, we have undertaken a review of the formatting and numbering of our Tables, ensuring readability and coherence throughout the manuscript. Every revision has been highlighted, for easy reference.

Once again, we express our gratitude for your guidance, which has undeniably elevated the quality and rigor of our research.

Round 3

Reviewer 2 Report

The authors provider the table comparing the study and control groups. First impression is that control group is not age and comorbidities matched to study one. I understand it is possible considering the type of symptom complained. 

To finally increase the value of the manuscript and make it suitable for publication I suggest the following upgrades

1- are the patients enrolled consecutive? If so please declare it in method. It could partially explain the differeces

2- is statistical analysis and correlation corrected for age and comorbidities? 

3- are controls afflicted by anxiety? 

4- due these differences may interfere with results, they must be declared in limitations

Author Response

We appreciate the insightful suggestions provided by the reviewer. The reviewer has highlighted the disparities between the patient and control groups in our study, which stem from the context in which patient recruitment was conducted – the emergency department of a specialized cardiovascular care center. This context inherently introduces a bias towards patients presenting with acute cardiovascular conditions, resulting in a higher prevalence of STEMI compared to cases of non-ischemic chest discomfort.

In response to the reviewer's recommendations, we have taken steps to address this concern. We have incorporated the following statement into the study design section: "Patients were consecutively recruited and were eligible for inclusion if they were admitted to the ED referring symptoms suggestive of an ACS..." This clarification helps contextualize the disparities in patient characteristics and reinforces the rationale for the observed differences between the study groups.

Additionally, we acknowledge the limitations arising from the disparities in age and comorbidity prevalence between the patient group and the control cohort. This potential source of confounding has been added as a limitation in the study, stated as follows: "Fourthly, disparities in age and the prevalence of comorbidities between the patient group and the control individuals could introduce confounding variables that may influence the observed results".

Regarding the reviewer's suggestion for statistical analyses adjusted for age and comorbidities, we concur with the need for caution in interpreting such analyses. The study was not originally designed for comprehensive assessment of differences in lncRNA expression across distinct age strata or specific underlying diseases. As aptly pointed out, the limited number of controls with hypertension or diabetes precludes meaningful multivariate analyses in this context.

Lastly, we appreciate the reviewer's clarification regarding the control group's health status. Notably, none of the control subjects exhibited underlying conditions characterized by anxiety, and only one individual was identified as having experienced a panic attack. The prevailing diagnoses among the control group primarily comprised cases of costochondritis and non-specific musculoskeletal pain. This has been added in the "Study population" section.

We are grateful for the constructive feedback provided by the reviewer, which has enriched the quality and accuracy of our study's presentation. The aforementioned modifications have been implemented to enhance the transparency and interpretation of our findings.

Round 4

Reviewer 2 Report

The authors did a great job in revising the manuscript. Now I think is ready for publication

Author Response

I wish to express my sincere gratitude for the invaluable time and effort you have invested in meticulously reviewing and editing my manuscript.